# The Association between Sleep Quality and Fatigue in Colorectal Cancer Survivors up until Two Years after Treatment: A Cross-Sectional and Longitudinal Analysis

**DOI:** 10.3390/cancers14061527

**Published:** 2022-03-16

**Authors:** Meera Legg, Ree M. Meertens, Eline van Roekel, Stéphanie O. Breukink, Maryska L. Janssen, Eric T. P. Keulen, Karen Steindorf, Matty P. Weijenberg, Martijn Bours

**Affiliations:** 1Department of Health Promotion, Care and Public Health Research Institute (CAPHRI), School of Nutrition and Translational Research in Metabolism (NUTRIM), Maastricht University, 6200 MD Maastricht, The Netherlands; meeralegg@hotmail.co.uk; 2Department of Epidemiology, GROW School for Oncology and Developmental Biology, Maastricht University, 6200 MD Maastricht, The Netherlands; eline.vanroekel@maastrichtuniversity.nl (E.v.R.); mp.weijenberg@maastrichtuniversity.nl (M.P.W.); m.bours@maastrichtuniversity.nl (M.B.); 3Department of Surgery, GROW School for Oncology and Developmental Biology, Maastricht University Medical Center+, 6202 AZ Maastricht, The Netherlands; s.breukink@mumc.nl; 4Department of Clinical Epidemiology, VieCuri Medical Centre, Tegelseweg 210, 5912 BL Venlo, The Netherlands; maryska.janssen@maastrichtuniversity.nl; 5Department of Internal Medicine and Gastroenterology, Zuyderland Medical Centre, Dr. H. van der Hoffplein 1, 6162 BG Sittard-Geleen, The Netherlands; e.keulen@zuyderland.nl; 6Division of Physical Activity, Prevention and Cancer, National Center for Tumor Diseases (NCT) and German Cancer Research Center (DKFZ), Im Neuenheimer Feld 280, 69120 Heidelberg, Germany; k.steindorf@dkfz-heidelberg.de

**Keywords:** colorectal, cancer, fatigue, sleep, insomnia, patients, quality of life

## Abstract

**Simple Summary:**

Fatigue is a distressing complaint with high detriment to quality of life that persists in one third of colorectal cancer survivors after cancer treatment. Surprisingly, the contribution of poor sleep quality to fatigue in colorectal cancer survivors is underinvestigated. We aimed to investigate the association between sleep quality and fatigue in colorectal cancer survivors up until two years post-treatment. Results showed worse sleep quality in colorectal cancer patients was associated with higher levels of fatigue during the first two years post-treatment. The results of this study suggest that more attention for sleep quality in colorectal cancer survivors and offering sleep health interventions may lead to less fatigue and better quality of life in this group.

**Abstract:**

Fatigue is a distressing complaint with high detriment to quality of life that persists in one-third of colorectal cancer survivors after cancer treatment. Previous studies in mixed groups of cancer patients have suggested sleep quality is associated with fatigue. We aimed to investigate this association in colorectal cancer survivors up until two years post-treatment. Data on *n* = 388 stage I–III colorectal cancer patients were utilized from the EnCoRe study. Sleep quality and fatigue were measured at 6 weeks and 6, 12, and 24 months post-treatment. Sleep quality was measured using the Pittsburgh Sleep Quality Index (cross-sectional analysis only) and the single-item insomnia scale from the EORTC QLQ-C30. Fatigue was measured by the Checklist Individual Strength. Linear and mixed-model regression analyses analysed associations between sleep quality and fatigue cross-sectionally and longitudinally. Longitudinal analysis revealed worsening sleep quality over time was significantly associated with increased levels of fatigue over time (β per 0.5 SD increase in the EORTC-insomnia score = 2.56, 95% Cl: 1.91, 3.22). Significant cross-sectional associations were observed between worse sleep quality and higher levels of fatigue at all time points. Worse sleep quality in colorectal cancer patients was associated with higher levels of fatigue during the first two years post-treatment.

## 1. Introduction

With 1.8 million new cases worldwide in 2018 [1], colorectal cancer (CRC) is the third most prevalent cancer in men and the second in women [1]. The mortality rates of CRC have recently been decreasing in high-income countries as a result of new or improved screening programs and advancements in cancer treatment [2]. Consequentially, the overall five-year CRC survival rate in Europe has more than doubled over the last 40 years, reaching 56% at present [2]. The growing population of individuals living with a past or present CRC diagnosis (CRC survivors) requires the development and distribution of sufficient supportive care to meet physical, social, and informational needs.

CRC survivors tend to experience many cancer-related symptoms that have a profound impact on health-related quality of life (HRQoL) [2]. These survivors have consistently reported persistent fatigue to be the most distressing and detrimental symptom to HRQoL as it restricts the engagement in everyday roles and activities that make life meaningful and often leads to changes in employment status (75% of CRC survivors), which can create a significant financial burden [3,4]. The National Comprehensive Cancer Network (NCCN) defines cancer-related fatigue as “a distressing, persistent, subjective sense of physical, emotional, and/or cognitive tiredness or exhaustion related to cancer or cancer treatment that is not proportional to recent activity and significantly interferes with usual functioning”.

The average prevalence of persistent fatigue in both short- and long-term CRC survivors (39%) is notably greater than the prevalence of fatigue in the normative Dutch population (21%) [5]. Fatigue severity is generally found to peak during cancer treatment and decline post-treatment [6], whereas fatigue prevalence varies throughout the course of cancer treatment and recovery, with 35% of CRC survivors continuing to experiencing fatigue more than five years after cancer diagnosis [7]. Despite the high prevalence and ramifications of fatigue after cancer, it is underreported and inadequately managed [3,4]. To improve its management and consequentially HRQoL in CRC survivors, researchers need to identify the causes of fatigue after cancer, which are not yet fully understood [4,7]. Since fatigue is a subjective, complex, and multidimensional construct, there are many potential contributing factors including biochemical factors such as chemotherapy, behavioural factors such as napping, and psychological factors such as the presence of depression and anxiety [7,8].

Sleep quality has also been proposed as a biological and psychological predictor of cancer-related fatigue, as poor sleep quality in non-cancer patients has been shown to contribute to increased fatigue [9]. Sleep disturbance is highly prevalent in cancer patients, ranging from 30% to 93% [10]. Cancer patients experience a wide range of sleep disturbances such as difficulty falling asleep, difficulty staying asleep, and non-restorative sleep [11].

Several cross-sectional studies have investigated the association between sleep quality and fatigue in the context of cancer (mainly breast cancer). The majority of these studies found worse sleep quality to be significantly associated with increased fatigue before [12], during [13,14,15,16], and after treatment [6,17]. However, a few cross-sectional studies found no association between sleep quality and fatigue during treatment [18,19] and in the post-treatment period [20]. To our knowledge, only two studies have investigated the cross-sectional association between sleep quality and fatigue in CRC survivors [21,22]. Both of these study populations involved CRC survivors who were in the treatment and post-treatment phase and found sleep quality to be significantly associated with fatigue [21,22].

Only two studies have investigated the longitudinal association between sleep quality and fatigue in cancer patients. One study found poorer sleep quality to be related to increased fatigue during chemotherapy [23], whereas the other found no significant association during radiotherapy [19]. To the best of our knowledge, no studies have investigated the longitudinal association between sleep quality and fatigue in CRC survivors during the post-treatment period. Therefore, the present study aims to investigate the longitudinal association between sleep quality and fatigue in CRC survivors from the period of six weeks post-treatment until two years post-treatment. We hypothesized that poorer sleep quality is associated with worse fatigue over time.

## 2. Materials and Methods

### 2.1. Participants and Study Design

Data were utilized from the ongoing Energy for life after ColoRectal cancer (EnCoRe) study (Netherlands Trial Register no. NL6904) [24]. The EnCoRe study is a prospective cohort study, recruiting CRC survivors at the time of cancer diagnosis from three participating hospitals in the Netherlands: Maastricht University Medical Center+, VieCuri Medical Center, and Zuyderland Medical Center. Recruitment commenced in April 2012. Eligible participants, identified and enrolled by nurse practitioners, were individuals over the age of 18 years with a diagnosis of stage I–III CRC. Participants were excluded if they did not understand Dutch, did not live in the Netherlands, or had co-morbidities that hindered participation. The EnCoRe study was approved by the medical ethics review committee of Maastricht University Hospital and Maastricht University. All participants provided written informed consent.

### 2.2. Data Collection Procedures

Data collection was undertaken during home visits. Demographic information was collected around the time of diagnosis, whilst other variables were collected at four post-treatment time-points: 6 weeks, 6 months, 12 months, and 24 months post-treatment. The number of participants who provided data at each time-point can be seen in Figure 1. Figure A1 in Appendix A additionally shows response rates and reasons for non-participation at each time-point.

Data up until July 2018 were used in the present analyses. The declining number of participants at subsequent time-points (Figure 1 and Figure A1 in Appendix A) is largely due to individuals not having reached the later time-points up until July 2018. Post-treatment loss to follow-up was less than 10% at each consecutive time-point.

### 2.3. Data Collection

Two measures of sleep quality were used in this study: the official Dutch version of the Pittsburgh Sleep Quality Index (PSQI, https://www.sleep.pitt.edu/instruments/#psqi) (accessed on 1 February 2022) and the insomnia scale of the official Dutch version of the European Organization for Research and Treatment of Cancer Quality of Life Questionnaire-Core 30 (EORTC QLQ-C30, https://qol.eortc.org/translations/) (accessed on 1 February 2022). The PSQI questionnaire is a validated 24-item subjective measure of sleep quality, consisting of seven components, all used to calculate a global score. The global PSQI score, which represents sleep quality over the past month, ranges from 0–21 points with higher scores indicating poorer sleep quality [25]. The PSQI has a high internal consistency (Cronbach’s *α* > 0.8) in many clinical populations [26] including cancer patients [27], see also [28,29]. In the current paper, poor sleep quality was defined as a PSQI score of five or greater, which has very good sensitivity (89.6%) and specificity (86.5%) for distinguishing people with sleep impairments [25].

The EORTC QLQ-C30 is a multidimensional, cancer-specific questionnaire consisting of 30 items assessing HRQoL [30], also validated in Dutch [31]. The questionnaire measures insomnia using the single item: “During the past week, have you had trouble sleeping?”, with four answer options (not at all, a little, quite a bit, very much), which are converted into a 100-point scale as per the questionnaire manual. Higher scores indicate worse insomnia [30]. Although no guidance for cut-off scores currently exists for the single-item insomnia scale, we considered participants reporting “a little”, “quite a bit”, or “very much” on this item as having poor sleep quality and those reporting “not at all” as having good sleep quality.

The PSQI was added to the measurements taken in the EnCoRe study from August 2017 onwards. As a result, the number of participants who provided data on sleep quality through the PSQI questionnaire was smaller in comparison to sleep quality as measured by the single-item EORTC insomnia scale (see Figure 1), which was included from the start of the EnCoRe study in 2012.

Fatigue was measured by the Checklist Individual Strength (CIS), a questionnaire originally developed and validated in Dutch [32] to assess fatigue in patients with chronic fatigue syndrome, but is also commonly used to assess fatigue in cancer survivors [33,34]. This questionnaire contains 20 items, each using seven-point Likert response scales. A global fatigue score, calculated based on all items, ranges from 20–140 points (higher scores representing worse fatigue) and indicates average fatigue levels over the past week. The CIS is comprised of four subscales, including subjective fatigue (eight items), concentration problems (five items), reduced motivation (four items), and reduction of activity (three items). The CIS has a high internal consistency (Cronbach’s *α* = 0.84–0.95) and good test–retest reliability (*r* = 0.74–0.86) [33]. A cut-off score of 35 or higher on the subjective fatigue subscale was employed to identify participants with severe fatigue and 27–34 for elevated fatigue [35].

Sociodemographic characteristics including age and gender were self-reported by participants at diagnosis. Clinical characteristics including tumor site, cancer stage, and treatment with chemotherapy and/or radiotherapy were collected from clinical records. Additionally, levels of anxiety and depression were assessed at post-treatment time-points via the Dutch version of the Hospital Anxiety and Depression Scale (HADS), which has good construct validity and reliability in cancer patients [36] and also has been validated in Dutch [37]. A cut-off of eight points or higher on the anxiety and depression subscales (0–21 points) of the HADS indicates the presence of considerable symptoms of anxiety or depression [36].

### 2.4. Statistical Analysis

Disease and demographic characteristics were described for all participants involved in this study. The means and standard deviations of sleep quality (as measured by both PSQI and EORTC QLQ-C30) and fatigue were calculated as well as the proportion of individuals with poor sleep quality and moderate to severe fatigue at each time-point. An independent T-test was conducted to detect significant changes in the global PSQI scores between 6 months and 24 months post-treatment (the PSQI measurements at 6-month and 24-month follow-up time-points were not overlapping, meaning that there were no participants who filled in the PSQI at both time-points because the PSQI was added to the study measurements 5 years after the start of the study). Next, mixed models including only ‘time since end of treatment’ as a covariate were used to analyse how insomnia scores and how fatigue scores changed over time.

To test the cross-sectional association between sleep quality (PSQI) (independent variable) and fatigue (dependent variable), univariable and multivariable linear regressions were conducted at each time-point. Model 1 tested the unadjusted (crude) association. Confounders identified through literature were sequentially added in three additional hierarchically adjusted models. Identified confounders included age [38,39], gender [40,41], treatment with chemotherapy and/or radiotherapy [39,42], and time since end of treatment. Model 2 was established by adding age, gender, and time since end of treatment to Model 1. Model 3 additionally incorporated treatment with chemotherapy and/or radiotherapy as covariates. Model 3 was considered our final model in the current analyses. In an additional exploratory model, the HADS total score was added to Model 3 as a covariate to explore potential confounding by levels of psychological distress. This fourth model was considered exploratory as it is unknown whether anxiety and depression are confounders [43,44,45] or mediators [46] in the association between sleep quality and fatigue. Linear regression was similarly conducted to analyse associations between sleep quality (EORTC) and fatigue at each time-point, using the same stepwise addition of covariates. The insomnia variable was rescaled prior to analyses to aid interpretation of the magnitude of any association discovered between insomnia and fatigue. The original single-item EORTC insomnia scores on the 100-point scale were divided by 0.5× the standard deviation of insomnia scores at 6 weeks post-treatment. Thus, beta coefficients in these analyses represented the change in fatigue score per 0.5 SD insomnia score, representing clinically relevant increments in insomnia. This rescaled variable was utilized in all further analyses. Assumptions of the regression were assessed and met in all models.

Next, linear mixed models were utilized to analyse the longitudinal association between insomnia and fatigue from 6 weeks until 2 years post-treatment. PSQI sleep quality data were not used in this analysis due to insufficient quantities of follow-up data. Mixed-model regression techniques use random intercepts to account for the intra-individual dependency between repeated measures over time. Mixed-model regression is a well-established longitudinal analysis technique that enables use of all collected data (including data from participants with missing values at one or more time-points), thereby minimizing data loss and optimizing power [47]. Moreover, these models allow for disaggregation of intra-individual from inter-individual associations over time in order to explore whether associations between time-dependent exposures and outcomes involve within-person changes over time and/or between-person differences over time [48,49].

Thus, mixed models were selected for the longitudinal analyses, since they include both fixed and random effects, account for correlated data within participants, and handle missing data well. Confounders for the mixed model analyses were added in the same stepwise fashion as the aforementioned multivariable regression analyses.

Linear mixed modeling provides a beta coefficient, which is a weighted average of the between-subject component and within-subject component. The within-subject component represents how changes in insomnia within individuals are related to fatigue over time, while the between-subject component reflects how average differences in insomnia between subjects over time are longitudinally related to fatigue [50]. To disaggregate the within-subject and between-subject components, hybrid modeling was implemented [50]. Here, the between-subject component was modeled as each participant’s mean insomnia score across all time-points, and the within-subject component was modeled as the deviation score for each participant at every time-point, from an individual’s mean insomnia score across all time-points.

In addition, a time-lag analysis using linear mixed models was performed to explore the directionality of the association between insomnia and fatigue over time, by modeling how insomnia at the 6-week, 6-month, and 12-month time-points was related to fatigue at the 6-month, 12-month, and 24-month time-points, respectively. Hybrid modeling was again implemented to disentangle within-subject and between-subject components.

For all mixed model analyses, a random slope was tested for significance. No analyses found the addition of a random slope to be significant, and thus it was not included in the models. The assumptions of all mixed models were met; the residuals were normally distributed, linear, and had a constant variance. For all analyses, a *p* value less than 0.05 was considered statistically significant. Mixed model linear regression was conducted in Stata vs 16.1. All other analyses were conducted in SPSS version vs 26.

## 3. Results

### 3.1. Participant Characteristics

The participant characteristics can be seen in Table 1 (*n* = 389). The participants had a mean age at enrolment of 67 years (*SD* = 9.1) and the majority of participants were male (68%). Two-thirds of participants had colon cancer (64%) and one-third had rectal cancer (37%). Nearly half the participants were diagnosed with stage III cancer (43%), with less participants being diagnosed with stage I cancer (32%) and stage II cancer (25%). Most participants received surgery as treatment (90%); 39% had undergone chemotherapy and 25% radiotherapy. Anxiety was present in 37 participants (10%) and depression was present in 53 participants (14%) at 6 weeks post-treatment. These percentages and means for all disease and demographic characteristics were approximately the same in the overlapping samples: those who provided data on both PSQI sleep quality and fatigue at one or more time-points (*n* = 201) and those who provided data on both EORTC insomnia and fatigue at one or more time-points (*n* = 388; see Table A1 in Appendix B). The means and percentages also appeared similar between participants with poor sleep quality and participants with normal sleep quality. The only notable difference was higher percentages of anxiety and depression in those with poor sleep quality (13.7% and 20.2%, respectively), compared to participants with normal sleep quality (5.8% and 7.8%, respectively). Participants with poor sleep quality were also more often females (68% compared to 55%).

### 3.2. Trends over Time

Insomnia (EORTC) and sleep quality (PSQI) measures showed a high correlation at all time-points (6 weeks post-treatment: *r* = 0.81; 6 months: *r* = 0.74; 12 months: *r* = 0.75; and 24 months: *r* = 0.78). There was no significant difference in mean PSQI scores between the 74 participants at 6 weeks post-treatment (5.23 points, *SD* = 3.44) and the 71 participants at 24 months post-treatment (5.51 points, *SD* = 3.58; *t* (143) = −0.48, *p* = 0.67). The percentage of participants with poor sleep quality based on the PSQI cut-off score increased from 47.3% at six weeks post-treatment to 52.4% at 24 months post-treatment. The percentage of participants with moderate or severe fatigue decreased from 49.4% at six weeks post-treatment to 36.2% 24 months post-treatment. The linear mixed model analyses, with time since end of treatment as the independent variable, revealed that fatigue scores decreased significantly over time (*β* per 6 months = −1.91, 95% *Cl*: −2.54, −1.27), whereas EORTC insomnia scores did not significantly change over time (*β* per 6 months = −0.05, *Cl*: −0.11, 0.01). The mean scores and standard deviations of insomnia, sleep quality, and fatigue at each time-point can be found in Table A2 in Appendix C, showing that insomnia and sleep quality scores varied little over time, with a possible dip at 6 months.

### 3.3. Cross-Sectional Analyses

In the cross-sectional linear regression analyses (Table 2), poorer sleep quality (PSQI) was associated with worse fatigue at all post-treatment time-points. The final confounder-adjusted model (Model 3) showed a significant association at all time-points (6 weeks post-treatment: *β* per 1-point change of the PSQI score = 2.66, 95% *Cl*: 0.93, 4.39; 6 months: 2.96, 95% *Cl*: 1.56, 4.35; 12 months: *β* = 2.13, 95% *Cl*: 0.49, 3.78; and 24 months: *β* = 4.11, 95% *Cl*: 2.38, 5.83). Associations were attenuated and non-significant after additional adjustment for psychological distress in the exploratory model.

The cross-sectional linear regression analyses using insomnia (EORTC) as the independent variable (Table 2) also revealed that, after confounder adjustment (Model 3), poorer insomnia scores were associated with worse fatigue at all time-points (6 weeks post-treatment: *β* per 0.5 *SD* of insomnia scores = 4.50, 95% *Cl*: 3.25, 5.75; 6 months: *β* = 4.44, 95% *Cl*: 2.98, 5.90; 12 months: *β* = 5.13, 95% *Cl*: 3.55, 6.71; and 24 months: *β* = 6.06, 95% *Cl*: 4.26, 7.85). Associations were again attenuated in the exploratory model, which adjusted for psychological distress, although remaining statistically significant at the first two post-treatment time-points.

### 3.4. Longitudinal Analyses

Table 3 shows the results of the linear mixed model analyses. Higher insomnia scores over time were longitudinally associated with higher fatigue scores in the unadjusted model (Model 1: *β* = 2.71, 95% *Cl*: 2.07, 3.38), which was slightly attenuated after the addition of demographic and time covariates (Model 2: *β* = 2.58, 95% *Cl*: 1.93, 3.23) and treatment covariates (Model 3: *β* = 2.56, 95% *Cl*: 1.91, 3.22). The longitudinal association was further attenuated upon adding the psychological distress covariate, though remaining statistically significant (exploratory model: *β* = 1.20, 95% *Cl*: 0.61, 1.79).

Hybrid modeling demonstrated that the longitudinal association between insomnia and fatigue was mainly driven by between-person differences in insomnia over time. In the mixed models, the between-person associations were much larger than the within-person associations, although both components were statistically significant in all models (Table 3). This indicated that people with higher insomnia scores over time reported worse fatigue on average than persons with lower insomnia scores (Model 3, between: *β* = 6.65, 95% *Cl*: 5.33, 7.97), and that fatigue scores also increased in persons in whom insomnia scores increased over time, albeit to a lesser extent (Model 3, within: *β* = 1.42, 95% *Cl*: 0.68, 2.15).

### 3.5. Time-Lag Analysis

Time-lag analyses (Table 3) revealed a significant association between insomnia at earlier timepoints and fatigue at later timepoints in all models, including the final model (model 3: *β* = 2.04, 95% *Cl*: 1.26, 2.82). Hybrid modeling demonstrated the between-person associations to be significant in all models, including the final model (Model 3, between: *β* = 6.30, 95% *Cl*: 4.88, 7.73). However, within-person associations were not significant in the time-lag models (Model 3, within: *β* = 0.49, 95% *Cl*: −0.41, 1.39).

## 4. Discussion

This study aimed to explore the association between sleep quality and fatigue in CRC survivors, both cross-sectionally and longitudinally from six weeks after the end of cancer treatment until two years post-treatment. Approximately one-third of participants reported elevated or severe complaints of fatigue at two years post-treatment and approximately half of the CRC survivors reported poor sleep quality throughout the entire period up to 2 years post-treatment. This indicates that persistent fatigue and poor sleep quality are common problems experienced by CRC survivors in the first two years after the end of cancer treatment. Additionally, on average, fatigue decreased while sleep quality remained constant over time, during the post-treatment period.

As hypothesized, the current study found lower sleep quality to be cross-sectionally associated with worse fatigue at all post-treatment time-points. This is in line with previous findings in CRC survivors in the post-treatment period [21,22], as well as in other cancer patients during the post-treatment period [6,17] and treatment period [18,19]. Furthermore, longitudinal analyses revealed changes in insomnia over time were significantly associated with changes in fatigue from six weeks post treatment until 24 months post-treatment. Only two previous papers have reported on the longitudinal association between sleep quality and fatigue in cancer patients, both of which solely looked at the treatment period [19,23]. A study in prostate cancer survivors found no significant association [19], whereas a study of breast cancer survivors found a significant association between worse sleep quality and more fatigue during the treatment period [23]. Thus, the current study, which focused on the longitudinal association in the post-treatment period up to 2 years after the end of CRC treatment, presents novel findings, adding substantially to the pre-existing evidence base.

Psychological distress (a compound factor of anxiety and depression) was controlled for in exploratory analyses, because it is uncertain whether anxiety and depression confound [46,47,48] or mediate [49] the association between sleep quality and fatigue in (colorectal) cancer patients. These exploratory analyses revealed largely attenuated associations, suggesting that psychological distress affects the impact of sleep quality on fatigue. More research is necessary to disentangle whether psychological distress confounds or mediates effects of sleep quality on fatigue.

Although this study was observational in nature and thus could not demonstrate causality, a time-lag analysis was conducted to explore the directionality of the association between sleep quality and fatigue over time. The results suggest that higher insomnia scores at earlier timepoints are related to higher fatigue scores at later time points, although this association may be predominantly driven by between-person differences rather than within-person differences in insomnia. A conclusion about the direction of association cannot be drawn, and further research is needed. It is likely, however, that insomnia and fatigue are bi-directionally related, as suggested by previous research [51].

Our study has several limitations. Firstly, both measures of sleep quality were self-reported, which could have caused recall bias as respondents were asked to recall sleep from the prior month. In future research, sleep quality could be objectively measured with actigraphy, to increase the validity and reliability of results. Secondly, the single-item insomnia scale of the EORTC questionnaire is not an in-depth measure of sleep quality. However, a high correlation between insomnia (EORTC) and sleep quality (PSQI) at all timepoints (Pearson *r*: 0.74–0.81), suggests that the EORTC insomnia scale captures a similar construct as the PSQI. Since a limited number of participants provided PSQI sleep quality data, a restricted number of confounders could be added to each model to avoid over-adjusting. Although the main confounders based on literature were adjusted for, residual confounding may have occurred as relevant confounders, such as Body Mass Index, physical activity, and number of comorbidities, would have ideally been included in the models. Future research should ensure adjustment of all relevant confounds.

This study also had several strengths. Due to intensive contact with the participants during the post-treatment follow-up period (i.e., all study measurement were performed through home visits), participation rates remained high (>90%) at each time-point. Loss to follow-up (e.g., due to passing away) was below 10%. As a consequence, the probability of selection bias due to selective loss to follow-up was limited. However, we cannot exclude the possibility that there might have been selection bias upon inclusion of patients in our study (e.g., that the healthier patients were more likely to participate).

Furthermore, the results are probably generalizable to the whole Dutch CRC survivor population as the participant characteristics are similar to this population. For example, the average age at CRC diagnosis in the Netherlands is 69 years of age [52], and 63% of newly diagnosed CRC patients are male [53]. In the EnCoRe study, the average age at diagnosis is 67 years old and approximately 68% are male. Other strengths of the study include being prospective, the longitudinal design, and the large sample size used in the longitudinal analyses.

## 5. Conclusions

Poor sleep quality and fatigue are both highly prevalent and distressing symptoms experienced by CRC survivors in the years following cessation of cancer treatment. The post-treatment association between worse sleep quality and increased fatigue reported previously in studies of other cancer types was confirmed in this study. There was a significant association between worse sleep quality and worse fatigue from six weeks post-treatment until 24 months post-treatment. The current study has identified and fulfilled a gap in the research on factors contributing to fatigue after cancer treatment. The results underline the need for improvement in sleep quality by advancing and developing interventions (e.g., Cognitive Behavioural Therapy for Insomnia) to decrease complaints of fatigue, with the ultimate aim of improving HRQoL in CRC survivors.

## Figures and Tables

**Figure 1 cancers-14-01527-f001:**
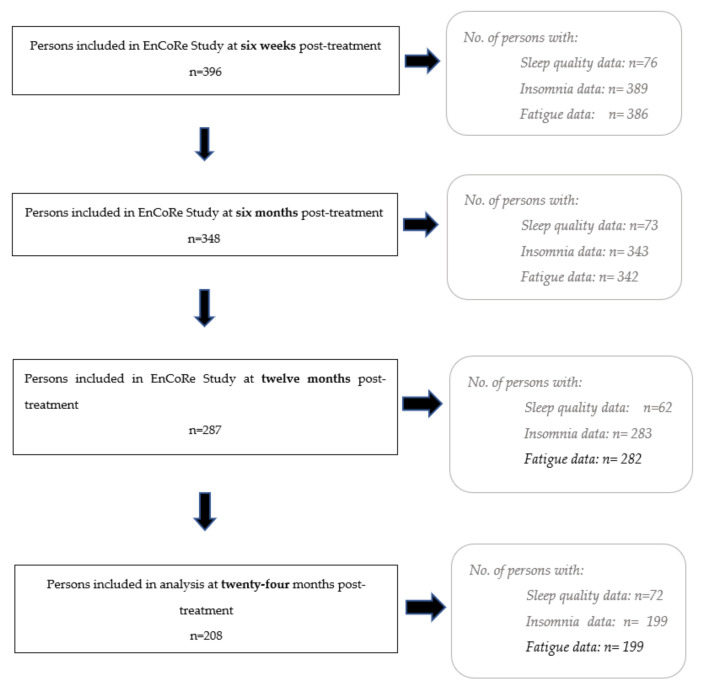
Flow diagram of the respondent frequency (*n*) at each time-point. 1. Data up until July 2018 were used in the present study. The declining number of participants at subsequent time-points is largely due to individuals not having reached the later time-points by July 2018. 2. The measurement of sleep quality was introduced to the study in 2017, five years after the start of the study. 3. Insomnia is a second measure of sleep quality used in this study 4. Not all participants included in the EnCoRe study at each timepoint provided data on sleep quality, insomnia, or fatigue. 5. In total, across all time-points, 201 persons provided data on both fatigue and sleep quality, whereas 388 participants provided data on both insomnia and fatigue. The participants who provided data on both variables may not have been the same at each timepoint.

**Table 1 cancers-14-01527-t001:** Demographic and clinical characteristics of participants.

	Total Population (*n* = 389)	Participants with Poor Sleep Quality ^a^ (*n* = 183)	Participants with Normal Sleep Quality ^b^ (*n* = 206)
Age at enrolment:			
Mean (*SD*)	66.6 (9.1)	66.3 (9.5)	66.9 (8.1)
min–max	36–90	40–90	36–88
Gender *n* (%):			
Male	266 (68.4)	115 (62.8)	151 (73.3)
Female	123 (31.6)	68 (37.2)	55 (26.7)
Time since end of treatment at 6 weeks post-treatment:			

Mean (*SD*)	7.95 (3.5)	7.82 (3.4)	8.07 (3.6)
min–max	3.3–37.3	3.9–37.3	3.3–32.3
Cancer location *n* (%):			
Colon	247 (63.5)	110 (60.1)	137 (66.5)
Rectum	142 (36.5)	73 (39.9)	69 (33.6)
Cancer Stage *n* (%):			
I	123 (31.6)	55 (30.1)	68 (33.0)
II	97 (24.9)	44 (24.0)	53 (25.7)
III	169 (43.4)	84 (45.9)	85 (41.3)
Treatment undergone *n* (%):			
Surgery	348 (89.5)	169 (92.4)	179 (87.0)
Chemotherapy	150 (38.6)	69 (37.7)	81 (39.3)
Radiotherapy	98 (25.2)	51 (27.9)	47 (22.8)
No treatment	18 (4.6)	4 (2.2)	14 (6.8)
Anxiety at 6 weeks post-treatment *n* (%) *:			
Present	37 (9.5)	25 (13.7)	12 (5.8)
Absent	351 (90.2)	157 (85.9)	194 (94.2)
Depression at 6 weeks post-treatment *n* (%) *:			
Present	53 (13.6)	37 (20.2)	16 (7.8)
Absent	335 (86.1)	145 (79.2)	190 (92.2)

^a^ as defined by the answers of “a little”, “often”, and “always” to the single item insomnia question (during the past week, have you had trouble sleeping) in the EORTC questionnaire. ^b^ as defined by the answer of “not at all” to the single item insomnia question (during the past week, have you had trouble sleeping) in the EORTC questionnaire. * One person did not provide data on anxiety or depression in the total population (*n* = 389) and in the population of participants with poor sleep quality (*n* = 183).

**Table 2 cancers-14-01527-t002:** Multiple linear regression results of sleep quality with fatigue and insomnia with fatigue at week 6, month 6, month 12, and month 24.

Regression	Model	Week Six ^1^	Month Six	Month Twelve	Month Twenty-Four								
		β ^2^	95% Cl	*p* Value	β	95% Cl	*p* Value	β	95% Cl	*p* Value	β	95% Cl	*p* Value
**Sleep Quality and Fatigue ^3^**	Model 1 ^a^	2.33	0.69, 3.97	0.006	2.83	1.37, 4.30	<0.001	2.48	0.85, 4.11	0.003	3.69	2.01, 5.37	<0.001
	Model 2 ^b^	2.53	0.86, 4.19	0.003	2.95	1.59, 4.30	<0.001	2.15	0.56, 3.74	0.008	4.11	2.44, 5.78	<0.001
	Model 3 ^c^	2.66	0.93, 4.39	0.003	2.96	1.56, 4.35	<0.001	2.13	0.49, 3.78	0.012	4.11	2.38, 5.83	<0.001
	Exploratory model ^d^	0.82	−0.90, 2.54	0.345	1.18	−0.18, 2.54	0.088	0.94	−0.55, 2.44	0.211	1.46	−0.06, 2.98	0.060
**Insomnia and Fatigue ^3^**	Model 1 ^e^	4.66	3.42, 5.90	<0.001	4.56	3.16, 5.95	<0.001	5.06	3.56, 6.56	<0.001	5.72	3.95, 7.49	<0.001
	Model 2 ^f^	4.56	3.35, 5.85	<0.001	4.52	3.07, 5.97	<0.001	5.19	3.62, 6.76	<0.001	6.06	4.26, 7.85	<0.001
	Model 3 ^g^	4.50	3.25, 5.75	<0.001	4.44	2.98, 5.90	<0.001	5.13	3.55, 6.71	<0.001	6.06	4.26, 7.85	<0.001
	Exploratory model ^h^	1.64	0.53, 2.75	0.004	1.21	0.01, 2.41	0.047	1.29	−0.10, 2.67	0.068	1.58	−0.01, 3.18	0.052

^a^ Sleep quality. ^b^ Sleep quality, time since end of treatment, gender, age. ^c^ Sleep quality, time since end of treatment, gender, age, chemotherapy (yes/no), radiotherapy (yes/no). ^d^ Sleep quality, time since end of treatment, gender, age, chemotherapy (yes/no), radiotherapy (yes/no), psychological distress. ^e^ Insomnia. ^f^ Insomnia, time since end of treatment, gender, age. ^g^ Insomnia, time since end of treatment, gender, age, chemotherapy (yes/no), radiotherapy (yes/no). ^h^ Insomnia, time since end of treatment, gender, age, chemotherapy (yes/no), radiotherapy (yes/no), psychological distress. ^1^ All beta coefficients in the top of the table represent the association between fatigue scores (dependent variable) per 1-point change of the sleep quality score (independent variable). ^2^ All beta coefficients in the bottom half of the table represent the association between fatigue scores (dependent variable) per ½ SD of insomnia scores (independent variable). ^3^ the number of CRC survivors involved in the insomnia and fatigue regression analyses at each timepoint: W6 (*n* = 386) M6 (*n* = 343), M12 (*n* = 283), M24 (*n* = 199). The number of CRC survivors involved in the sleep quality and fatigue regression analyses at each timepoint: W6 (*n* = 76), M6 (*n* = 73), M12 (*n* = 62), M24 (*n* = 72).

**Table 3 cancers-14-01527-t003:** Results of linear mixed models analyses and time-lag analyses.

Model	Components (within/between)	Linear Mixed Model Analysis				Results of Time-Lag Analysis			
		β Coefficient ^1^	95% Cl		*p* Value	β Coefficient	95% Cl		*p* Value
Model 1 ^a^	Overall model	2.72	2.07	3.38	<0.001	2.08	1.31	2.84	<0.001
	Within	1.54	0.80	2.29	<0.001	0.47	−0.42	1.37	0.297
	Between	6.68	5.36	7.99	<0.001	6.36	4.95	7.78	<0.001
Model 2 ^b^	Overall model	2.58	1.93	3.23	<0.001	2.07	1.29	2.85	<0.001
	Within	1.42	0.68	2.15	<0.001	0.48	−0.42	1.38	0.293
	Between	6.80	5.45	8.14	<0.001	6.54	5.08	8.00	<0.001
Model 3 ^c^	Overall model	2.56	1.91	3.22	<0.001	2.04	1.26	2.82	<0.001
	Within	1.42	0.68	2.15	<0.001	0.49	−0.41	1.39	0.285
	Between	6.65	5.33	7.97	<0.001	6.30	4.88	7.73	<0.001
Exploratory model ^d^	Overall model	1.20	0.61	1.79	<0.001	1.46	0.71	2.21	<0.001
	Within	0.69	0.02	1.36	0.044	0.49	−0.40	1.39	0.282
	Between	2.67	1.54	3.79	<0.001	3.76	2.39	5.13	<0.001

^a^ adjusted for insomnia. ^b^ adjusted for insomnia, time since end of treatment, gender, age. ^c^ adjusted for insomnia, time since end of treatment, gender, age, chemotherapy, radiotherapy. ^d^ adjusted for insomnia, time since end of treatment, gender, age, chemotherapy, radiotherapy, HADS total. ^1^ All beta coefficients represent the longitudinal association between fatigue scores (dependent variable) per ½ SD of the insomnia score (independent variable).

## Data Availability

Data described in the manuscript, code book, and analytic code will be made available upon request pending (e.g., application and approval, payment, other). Requests for data of the EnCoRe study can be sent to Martijn Bours, Department of Epidemiology, GROW-School for Oncology and Developmental Biology, Maastricht University, the Netherlands (email: m.bours@maastrichtuniversity.nl).

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
