# Peer review of "The Association between Sleep Quality and Fatigue in Colorectal Cancer Survivors up until Two Years after Treatment: A Cross-Sectional and Longitudinal Analysis"

_cancers, 2022, doi:10.3390/cancers14061527_

Round 1

Reviewer 1 Report

The authors used a sample of 396 individuals from the prospective population-based EnCoRe cohort to analyze the relationships between sleep quality and fatigue in colorectal cancer (CRC) survivors. Data from individuals who participated from 2012 and who were followed up to 2018 were used. The instruments used were the Checklist Individual Strength (CIS) to measure fatigue and the single item “insomnia” of the EORTC QLQ-C30 initially proposed to the participants. The Pittsburgh Sleep Quality Index (PSQI) was introduced secondarily (2017) which led to only a few patients having data.

The study is well designed and the manuscript is well written. Besides the limited number of survivors involved in the study, 52% of survivors were lost 24 months after treatment, a figure that excludes any extrapolation to Dutch CRC survivors. However, the authors recognize their work as exploratory, a qualifier that should be read in the title.  

Comments:

  1. Up to 2018, 4559 patients were enrolled of which 3462 (76%) had stage I-III disease (Sci Rep 2021;11:3923). Why were only 1099 survivors invited to participate (Fig. A1)?
  2. Figure 1: Should the reader understand that the 287 12-month survivors have data on insomnia?
  3. Statistical analysis (L224-227): Please explain the non-overlapping of the 6 month and 24 month populations?
  4. Statistical analysis: The number of participants decreased over time, introducing a bias in the longitudinal analysis. How missing data was handled in the analysis is crucial, especially with subjective data. Please comment.
  5. Statistical analysis: How many survivors have complete (4 time points) CIS and EORTC QLQ-C30 data? Could you perform a longitudinal analysis on this selected population? Results may be reported in Supplementary Data.
  6. Discussion (L451-453): The data reported are results not given in the results section of the manuscript. Please correct.
  7. Discussion (L456): Spell BMI in full.
  8. Discussion (L464-465): I do not agree with the qualifier “small”.
  9. Table 2: Change the header of the 1rst column to β.
  10. Table 3: Indicate the number of survivors included in the two models.

Author Response

We thank the reviewer for the compliments on the design and on how the manuscript is written, and the constructive comments. Below, we react to the comments of the reviewer.

General comments:

  • The overview the reviewer gives of the manuscript are ended with the following general comment: ‘Besides the limited number of survivors involved in the study, 52% of survivors were lost 24 months after treatment, a figure that excludes any extrapolation to Dutch CRC survivors. However, the authors recognize their work as exploratory, a qualifier that should be read in the title.’ 

*Reaction: It is a misunderstanding that 52% of the survivors were lost to follow-up at the 24-month post-treatment time-point. The dataset that was used for the analyses included data from participants collected up until July 2018. At that time, not all participants had reached all follow-up moments in our study. Consequently, missing data from these participants is not due to loss to follow-up. The participation rate during the follow-up measurements was actually above 90%. As stated in the text (page 3, L 135-138 of original manuscript), ‘The declining number of participants at subsequent time-points (Figure 1 and Figure A1 in Appendix A) is largely due to individuals not having reached the later time-points up until July 2018. Post-treatment loss to follow-up was less than 10% at each consecutive time-point.’  Although we provided this explanation in the methods section of the manuscript, we realize (with hindsight) that this might not have been clear enough. Therefore, we now address this issue also in the discussion: ‘Due to intensive contact with the participants during the post-treatment follow-up period (i.e. all study measurement were performed through home visits), participation rates remained high (>90%) at each time-point. Loss to follow-up (e.g. due to passing away) was below 10%. As a consequence, the probability of selection bias due to selective loss to follow-up was limited. However, we cannot exclude the possibility that there might have been selection bias upon inclusion of patients in our study (e.g. that the more healthy patients were more likely to participate).’

Specific comments

1. Reviewer: Up to 2018, 4559 patients were enrolled of which 3462 (76%) had stage I-III disease (Sci Rep 2021;11:3923). Why were only 1099 survivors invited to participate (Fig. A1)?

 *Reaction: The reviewer is referring to a study that used another Dutch cohort, i.e. the PLCRC cohort, which recruits patients from multiple centers in the Netherlands. The numbers mentioned by the reviewer (4559 and 3462) apply to this cohort. The participants included in the analyses in our manuscript came from another cohort study, the EnCoRe study. The EnCoRe study recruits participants in three centers, which is less than the number of recruitment centers involved in the PLCRC cohort. Another recent publication in Scientific Reports includes a detailed flow chart, showing that 1099 survivors were invited for participation in our study (Sci Rep 2021; 11:12440).

2. Reviewer: Figure 1: Should the reader understand that the 287 12-month survivors have data on insomnia?

*Reaction: No, but the reviewer is right that insomnia at 12 months was missing in the Figure. Thank you for noticing this mistake. We now added these data, and also added the data about fatigue to Figure 1. (See page 4 of revised manuscript; the figure now shows that 283 12-month survivors have data on insomnia).

3. Reviewer: Statistical analysis (L224-227): Please explain the non-overlapping of the 6 month and 24 month populations?

*Reaction: As mentioned in the text (L 195-199 original manuscript), the PSQI measurements were added to the study measurements in August 2017, which was five years after the start of the EnCoRe study. This was done for all study participants, regardless of the follow-up moment they had reached at that time (e.g. 6 or 24 months after treatment). As mentioned in our reply to the general comment, we used data collected up to July 2018 for the present analyses, at which time the PSQI had been included for less than a year. As a result, there were no participants who filled in the PSQI at the 6-month and the 24-month measurement in July 2018. Thus, the study populations at these time points were non-overlapping with regard to sleep quality assessed by the PSQI. We now explain this in the text (see page 5): ‘(the PSQI measurements at 6-month and 24-month follow-up time-points were not overlapping, meaning that there were no participants who filled in the PSQI at both time-points because the PSQI was added to the study measurements 5 years after the start of the study).’

4. Reviewer: Statistical analysis: The number of participants decreased over time, introducing a bias in the longitudinal analysis. How missing data was handled in the analysis is crucial, especially with subjective data. Please comment.

 *Reaction: First, we would like to stress once more (see our reaction to the general comment) that loss to follow-up was limited (<10%). As explained, the declining number of participants at subsequent time-points is largely due to individuals not having reached the later time-points in July 2018. Participation rates at each post-treatment follow-up moment were high (>90%). It is thus very unlikely that the decreasing numbers at participants over time have introduced bias. As mentioned before, we address this issue now in the discussion (see reply to general comment).

Furthermore, we used linear mixed-model regression techniques to perform our longitudinal analyses. A particular strength of mixed-model regression is that it handles missing data in a fundamentally other way than for example a repeated-measures ANOVA, in which each case with one or more missing data points is excluded. Mixed-model regression is a well-established longitudinal analysis technique that enables use of all collected data (including data from participants with missing values at one or more time-points), thereby minimizing data loss and optimizing power (Twisk, 2003).

To address this point, and also comment 5 and 10 of the reviewer (see below), we now have included a paragraph on the principles of mixed-model regression (see page 6 of revised manuscript):

‘Mixed-model regression techniques use random intercepts to account for the intra-individual dependency between repeated measures over time. Mixed-model regression is a well-established longitudinal analysis technique that enables use of all collected data (including data from participants with missing values at one or more time-points), thereby minimizing data loss and optimizing power [47]. Moreover, these models allow for disaggregation of intra-individual from inter-individual associations over time in order to explore whether associations between time-dependent exposures and outcomes involve within-person changes over time and/or between-person differences over time [48,49]’

5. Reviewer: Statistical analysis: How many survivors have complete (4 time points) CIS and EORTC QLQ-C30 data? Could you perform a longitudinal analysis on this selected population? Results may be reported in Supplementary Data.

*Reaction: As can be seen in Figure 1, 199 participants have reached the last time point (24 months post-treatment), and thus have complete CIS and QLQ-C30 data. However, in this paper we used a linear mixed-model regression to examine longitudinal associations between the main exposures and outcomes. As explained in our reply to point 4 (see above), mixed-model regression handles missing data very well, being a well-established longitudinal analysis technique that minimizes data loss and optimizes statistical power (Twisk, 2003). In addition, as also mentioned in our explanation of the reasons for the decreasing numbers over time (see reply to the general comment and point 4), the missing data from participants at later time-points is most probably completely random. Performing a complete-case analysis, as suggested by the reviewer, is therefore not recommended because it would compromise the statistical power of our analyses and unnecessarily exclude valuable data from study participants.

6. Reviewer: Discussion (L451-453): The data reported are results not given in the results section of the manuscript. Please correct.

*The correlations between PSQI and insomnia were also given in the results section, L 316-318 of the original manuscript: ‘Insomnia (EORTC) and sleep quality (PSQI) measures showed a high correlation at all time-points (6 weeks post-treatment: r=0.81; 6 months: r=0.74; 12 months: r=0.75; and 24 months: r=0.78).’

7. Reviewer: Discussion (L456): Spell BMI in full.

*We now spell BMI in full: Body Mass Index

8. Reviewer: Discussion (L464-465): I do not agree with the qualifier “small”.

*For this comment, we refer to our reaction to the general comment and comment 4.

9. Reviewer: Table 2: Change the header of the 1rst column to β.

 *We changed B to β in the header.

10. Reviewer: Table 3: Indicate the number of survivors included in the two models.

 *The numbers of survivors that are included in the models can be derived from the flowchart in Figure 1, as mixed-model regressions use all available data. This is now explained clearer in the text (see the text cited in our reaction to comment 4).

-Furthermore, we checked English language again, which led to some improvements and corrections.

Reviewer 2 Report

This study included 388 patients with stage 1-3 colorectal cancer from a, ongoing prospective study (EnCore study).

Sleep quality was assess using the Pittsburg Sleep Quality Index (PSQI), insomnia was evaluated with European Organization for Research and Treatment 143 of Cancer Quality of Life Questionnaire-Core 30 (EORT QLQ-C30), and fatigue was measured by the checklist individual strength.

Patients were evaluated at 6 weeks, 6 months, 12 months and 24 months post treatment.

Cross sectional was performed with univariable and multivariable linear regression at each time-point. Poor sleep quality was associated with worse fatigue at all post treatment point. Insomnia was also associated with worse fatigue at all time-points.

Longitudinal analysis was performed using linear mixed models to analyze longitudinal association between insomnia and fatigue from 6 weeks to 2 years post treatment. Higher insomnia scores were associated with higher fatigue scores.

The poor sleep quality based on PSQI increased from 47% at 6 weeks post treatment to 52% at 24 months post treatment.

The prevalence of moderate to severe fatigue decreased from 49% to 36%.

This study shows high prevalence of poor sleep quality and insomnia in patients with cancer.  Those with persistent fatigue were associated with worse sleep quality and worse insomnia.

Minor changes: please report if questionnaires were provided in English or Dutch, and if this was done in Dutch, please provide if there is a reference for study validation.

Author Response

We thank the reviewer for the positive reaction and constructive feedback, and react below to his/her comment:

Reviewer: ‘Minor changes: please report if questionnaires were provided in English or Dutch, and if this was done in Dutch, please provide if there is a reference for study validation.’

*Reaction: The questionnaires were provided in Dutch, as our study only included participants from the Netherlands. We used validated questionnaires. The CIS is a questionnaire that was originally developed and validated in Dutch (Vercoulen et al., 1994). For the PSQI, an official translation was used (https://www.sleep.pitt.edu/instruments/#psqi, see also for evidence for validity: Santosa et al., 2020). The EORTC is also available in a Dutch version (https://qol.eortc.org/translations/), and also a validation study is available (Stiggelbout et al., 2016). For the HADS, a Dutch validation study is available too (Spinhoven et al., 1997). We now added to the text that the questionnaires were in Dutch, and refer to the above-mentioned references (see pages 3-5).

-Furthermore, we checked English language again, which led to some improvements and corrections.

Round 2

Reviewer 1 Report

The authors have responded precisely to all the points raised.

The manuscript has improved a lot with the changes made.